# Incidence of Confirmed Influenza and Pneumococcal Infections and Vaccine Uptake Among Virologically Suppressed People Living with HIV

**DOI:** 10.3390/vaccines13040358

**Published:** 2025-03-27

**Authors:** Edith Wolder Ejlersen, Josefine Amalie Loft, Marco Gelpi, Safura-Luise Heidari, Omid Rezahosseini, Johan Runge Poulsen, Dina Leth Møller, Zitta Barrella Harboe, Thomas Benfield, Susanne Dam Nielsen, Andreas Dehlbæk Knudsen

**Affiliations:** 1Department of Infectious Diseases, Copenhagen University Hospital-Rigshospitalet, 2100 Copenhagen, Denmark; edith.wolder.ejlersen@regionh.dk (E.W.E.); josefine.amalie.loft.jakobsen@regionh.dk (J.A.L.); marco.gelpi@regionh.dk (M.G.); safura-luise.heidari.02@regionh.dk (S.-L.H.); johan.runge.poulsen.01@regionh.dk (J.R.P.); dina.leth.moeller@regionh.dk (D.L.M.); susanne.dam.poulsen@regionh.dk (S.D.N.); 2Department of Infectious Diseases, Copenhagen University Hospital–Amager and Hvidovre, 2650 Hvidovre, Denmark; thomas.lars.benfield@regionh.dk; 3Department of Pulmonary and Infectious Diseases, Copenhagen University Hospital, North Zealand, 3400 Hillerød, Denmark; omid.rezahosseini@regionh.dk (O.R.); zitta.barrella.harboe@regionh.dk (Z.B.H.); 4Department of Bacteria, Parasites, and Funghi, Statens Serum Institute, 2300 Copenhagen, Denmark; 5Department of Clinical Medicine, Faculty of Health and Medical Sciences, University of Copenhagen, 2200 Copenhagen, Denmark

**Keywords:** influenza, pneumococcal infections, *S. pneumoniae*, vaccine uptake, people living with HIV

## Abstract

Background/Objectives: Influenza and *Streptococcus pneumoniae* infections are common vaccine-preventable diseases to which people living with HIV (PLWH) may be more susceptible. This study aims to investigate the incidence of confirmed influenza and pneumococcal infections, and to determine the incidence rate (IR) and factors associated with vaccine uptake in a population of virologically suppressed PLWH. Methods: We included 1031 virologically suppressed PLWH from the Copenhagen Comorbidity in HIV Infection (COCOMO) study. Data on infections and vaccinations between 2015 and 2020 were collected from nationwide registries. Incidence rates with 95% confidence intervals (CIs) of confirmed influenza and pneumococcal infections and vaccine uptake were calculated, and predictors of vaccine uptake were explored using logistic regression. Results: The IR of influenza showed variation from year to year and ranged between 0 (95% CI: 0.0, 7.6) and 18.0 (95% CI: 8.2, 34.1) per 1000 person-years at risk with an overall IR of 8.4 per 1000 person-years at risk (95% CI: 5.4, 12.3). The overall IR of pneumococcal infections was 5.5 per 1000 person-years at risk (95% CI: 3.9, 7.5). Among PLWH, 53.2% were influenza-vaccinated at least once, 72.3% and 22.6% of PLWH were vaccinated at least twice and in all six seasons, respectively, while 31% had at least one pneumococcal vaccine. Previous pneumonia or bronchitis, higher body mass index, use of drugs to treat heart conditions, and longer time with HIV were independently associated with vaccine uptake. Conclusions: We found high incidences of confirmed influenza and pneumococcal infections in virologically suppressed PLWH, but vaccine uptake was below recommendations, highlighting the need for improved vaccination counseling.

## 1. Introduction

Since the initiation of combination antiretroviral therapy (cART), the incidence of pulmonary infections in people living with HIV (PLWH) has decreased [1,2,3]. Nonetheless, HIV infection remains a significant risk factor for hospital admissions due to pulmonary infections, even among PLWH on cART [3,4,5,6,7]. This elevated risk may be attributed to the higher prevalence of pulmonary comorbidities in PLWH [5,7,8,9,10]. Consequently, both the susceptibility to and morbidity from respiratory tract infections are likely to be higher in PLWH than in the general population.

Pneumonia is one of the most common infections among contemporary PLWH, with *Streptococcus pneumoniae* (*S. pneumoniae*) identified as a leading bacterial cause of pneumonia in developed countries [5,11,12,13]. Influenza is a very common viral infection that may lead to hospital admissions among PLWH [14,15,16,17]. Both invasive pneumococcal disease (IPD) and influenza are vaccine-preventable diseases, and vaccination against influenza virus and *S. pneumoniae* elicit protective antibody responses in PLWH [15,18,19,20], albeit vaccine-induced immune response in PLWH may vary according to immune status [19,21,22]. Thus, while some studies show consistent antibody responses regardless of CD4+ T lymphocyte counts at the time of vaccination, other studies suggest stronger antibody responses in PLWH with CD4+ T lymphocyte counts above 200 cells/µL or 500 cells/µL at the time of vaccination with the 23-valent pneumococcal polysaccharide vaccine, and above 350 cells/µL at the time of vaccination with the inactivated, quadrivalent influenza vaccine [13,19].

Vaccination against both influenza virus and *S. pneumoniae* is recommended for all PLWH, for *S. pneumoniae* preferably when the CD4+ T lymphocyte count is above 200 cells/µL [23,24]. Current vaccination guidelines recommend an annual influenza vaccine and at least one pneumococcal vaccine [23,24,25,26], and Danish guidelines specifically recommend vaccination of persons with immunodeficiency [27]. Despite this, low vaccine uptake has been reported, with a lack of guidance and perception of not needing immunization cited as primary reasons for non-vaccination [28,29]. These findings highlight the need for improved vaccination counseling and the need to identify PLWH who are not yet vaccinated.

In this study, we aimed to investigate the incidence rates (IRs) for confirmed influenza and pneumococcal infections, as well as the vaccine uptake and predictors of vaccine uptake in a population of virologically suppressed PLWH.

## 2. Materials and Methods

### 2.1. Study Setting

Participants were recruited from the Copenhagen Comorbidity in HIV Infection (COCOMO) study, which is an observational, prospective cohort study investigating the burden of non-AIDS comorbidity among PLWH [30]. Between March 2015 and December 2016, 1099 PLWH aged 20 years or older were included, representing more than 40% of PLWH residing in the Copenhagen area. In the present study, we included all PLWH with undetectable plasma viral load, defined as HIV RNA < 50 copies/mL (Figure A1).

At baseline, participants answered structured questionnaires to collect information concerning medical history and lifestyle, including smoking status and alcohol consumption. HIV-related characteristics, including blood CD4+ T lymphocyte count, plasma viral load, and use of cART were extracted from medical records. A physical examination, including height and weight, was performed by trained healthcare professionals using a standardized protocol [30]. 

Data on infection with the influenza virus and/or *S. pneumoniae* between the date of inclusion and 31 December 2021, were collected from the Danish Microbiology Database (MiBa) [31]. MiBa was established in 2010 and contains complete, nationwide microbiology data from Denmark, including results from general practitioners and hospitals. Infection with influenza was defined as a positive PCR for influenza A or B from a pharyngeal swap or sputum. Infection with *S. pneumoniae* was defined as a positive culture from sputum, paranasal sinuses, bronchoalveolar lavage, pleura, cerebrospinal fluid, joint fluid, or bacteremia. IPD was defined as a positive culture from cerebrospinal fluid, joint fluid, or bacteremia.

The type, date, and number of influenza and pneumococcal vaccines were obtained from the Danish Vaccination Registry (DDV) [32]. DDV is a national database containing all vaccinations performed in Denmark, with mandatory registration since 2015.

### 2.2. Statistical Analysis

Participants were followed from 15 March 2015 to migration, death, or 31 December 2021, whichever came first.

Influenza vaccine rates were calculated for each influenza season, running from 1 October to 31 March the following year, among participants who were alive and under follow-up at the end of each season. Incidence of influenza was calculated as the total number of events divided by cumulated person-years at risk of influenza, defined as the sum of population time at risk in influenza seasons. For each influenza season, time at risk was calculated from 1 October to influenza infection, death, migration, or 31 March of the following year, whichever came first. Pneumococcal vaccine rates were calculated using a competing risks approach, employing the cmprsk package in R [33]. This method accounts for the competing risks of migration and death when estimating the cumulative incidence function for receiving the pneumococcal vaccine. Pneumococcal infections were calculated as the total number of events divided by the total person-years of follow-up. As such, one individual may have had more cases of each infection. As the numbers of cases were low for both infections, we did not have the power to investigate factors associated with influenza and/or pneumococcal infection.

Danish guidelines recommend influenza and pneumococcal vaccination for individuals > 65 years of age, and individuals with chronic diseases, including acquired immune deficiencies [27,34]. Given these recommendations, we selected variables related to general health and chronic conditions, that may influence vaccine uptake. Specifically, factors such as age, sex, smoking status, alcohol use, body mass index (BMI), episodes of acute bronchitis or pneumonia in the past 10 years, use of drugs to treat heart conditions as well as HIV-specific variables, including time with HIV per five years, CD4+ T lymphocyte counts and CD4+ T lymphocyte nadir categorized in <200, 200–500 and >500 cells/µL, and previous AIDS-defining conditions, were explored using logistic regression. Statistical significance was defined as *p* < 0.05. Statistical analyses were conducted using R 2023.12.1+402.

### 2.3. Ethics

The COCOMO study (NCT02382822) has been approved by the Committee on Health Research Ethics of the Capital Region of Denmark (H-8-2024-004) and the Danish Data Protection Agency (protocol code: 30-1454, date of approval: 13 February 2015).

Oral and written informed consent was obtained from all participants.

## 3. Results

### 3.1. Study Participants

We included 1031 PLWH in the study with a total of 2995 person-years of follow-up in influenza seasons and 7308 person-years of follow-up for pneumococcal infections. Most participants were males (85%), with a median age of 51 years at the time of inclusion, and a mean BMI of 25.0 kg/m^2^. The vast majority (99%) were treated with cART, and the mean CD4+ T lymphocyte count was 724 cells/µL (SD 282). A total of 182 (18%) participants had a previous AIDS-defining condition. The baseline characteristics for the study population are summarized in Table 1.

### 3.2. Incidence of Infection

Between 2015 and 2020, the overall confirmed influenza IR was 8.4 per 1000 person-years (95% confidence interval, CI: 5.40, 12.32), with 25 cases in 25 participants.

The IRs of confirmed influenza each season between 2015 and 2020 are listed in Table 2.

None of the PLWH had more than one case of influenza in the study period. A total of 11/25 (44%) PLWH with confirmed influenza received therapy with oseltamivir.

The IR of pneumococcal infections was 5.5 per 1000 person-years (95% CI: 3.9, 7.5) based on 40 cases in 31 participants. Two PLWH experienced both influenza and pneumococcal infection in the study period. Of the 40 cases of positive pneumococcal samples, thirty-four (85%) were collected from sputum, four (10%) from bronchoalveolar lavage, and two (5%) from blood. There were no cases of *S. pneumoniae* in cerebrospinal fluid, joints, pleura, or sinus. This resulted in an IR of IPD of 0.3 per 1000 person-years (95% CI: 0.0, 1.0). Both cases of IPD occurred in 2017.

### 3.3. Vaccine Uptake

A total of 548 (53.2%) PLWH received at least one dose of influenza vaccine in the seasons from 2015 to 2020. Of these, 398 (72.3%) and 124 (22.6%) of PLWH were vaccinated twice and in all six seasons, respectively (Table 1). A total of 438 (46.9%) received no influenza vaccine in the study period. The incidence rates for influenza vaccination increased gradually for each consecutive season with a vaccination uptake of 213/1027 (20.7%) in 2015, 256/1023 (25.0%) in 2016, 289/1008 (28.7%) in 2017, 322/1000 (32.2%) in 2018, 327/990 (33%) in 2019, and 438/981 (44.6%) in 2020 (*p*, for trend < 0.001) (Figure 1).

Of the 1031 included participants, 320 (31%) received at least one pneumococcal vaccine between 2015 and 2021. Among PLWH that received at least one vaccine, 232 (72.5%) individuals received only the polysaccharide vaccine, 11 (3.4%) were vaccinated with the conjugate vaccine alone, and 77 (24.1%) received both pneumococcal vaccines. A total of 711 (69%) received no pneumococcal vaccine in the study period. Among PLWH that were vaccinated, 73.4% were vaccinated in 2020 (Table 1).

### 3.4. Factors Associated with Vaccine Uptake

Factors associated with vaccine uptake were assessed as odds ratio (OR) of receiving vaccination in an unadjusted model and in a model adjusted for age and sex (Figure 2 and Figure 3).

For the influenza vaccine, older age (OR per decade older 2.0, 95% CI: 1.8, 2.3), acute bronchitis or pneumonia in the past 10 years; 1–5 episodes (OR 1.9, 95% CI: 1.4, 2.5), 6–10 episodes (OR 3.2, 95% CI: 1.3, 8.2), and >10 episodes (OR 3.7, 95% CI: 1.3, 10.0) compared to no episodes, respectively, use of drugs to treat heart conditions (OR 5.1, 95% CI: 1.9, 13.3), increase in BMI per 1 kg/m^2^ (OR 1.04, 95% CI: 1.01, 1.07), former smoking (OR 1.7, 95% CI: 1.3, 2.4), CD4+ T lymphocyte nadir >500 cells/µL compared to <200 cells/µL, and longer time with HIV per five years (OR 1.22, 95% CI: 1.14, 1.32) were associated with higher odds of vaccine uptake. However, when adjusting for age and sex, only 1–5 episodes of acute bronchitis or pneumonia in the past 10 years (adjusted OR 1.6, 95% CI: 1.2, 2.2), use of drugs to treat heart conditions (adjusted OR 2.9, 95% CI: 1.1,8.0), and higher BMI (adjusted OR 1.04, 95% CI: 1.0, 1.07) remained significantly associated with higher odds of influenza vaccine uptake. No other investigated factors were associated with influenza vaccine uptake.

For the pneumococcal vaccines, older age (OR per decade older 3.5, 95% CI: 2.9, 4.2), acute bronchitis or pneumonia in the past 10 years; 1–5 episodes (OR 1.9, 95% CI: 1.4, 2.6), 6–10 episodes (OR 4.7, 95% CI: 2.0, 10.9), and >10 episodes (OR 4.9, 95% CI: 2.0, 11.9) compared to no episodes, respectively, former smoking (OR 1.7, 95% CI: 1.2, 2.4), use of drugs to treat heart conditions (OR 3.9, 95% CI: 1.9, 8.1), a previous AIDS defining condition (OR 1.6, 95% CI: 1.2, 2.3), CD4+ T lymphocyte nadir of 200–500 cells/µL (OR 0.70, 95% CI: 0.53, 0.92) and >500 cells/µL (OR 0.34, 95% CI: 0.20, 0.59) compared to <200 cells/µL, respectively, and longer time with HIV per five years (OR 1.50, 95% CI: 1.38, 1.62) were associated with higher odds of vaccine uptake. When adjusting for age and sex, only acute bronchitis or pneumonia in the past 10 years; 1–5 episodes (adjusted OR 1.60, 95% CI: 1.1, 2.3), 6–10 episodes (adjusted OR 4.4, 95% CI: 1.6, 11.6), and >10 episodes (adjusted OR 4.4, 95% CI: 1.6, 11.7), respectively, and longer time with HIV per five years (adjusted OR 1.16, 95% CI: 1.06, 1.28) were significantly associated with higher odds of vaccine uptake. No other investigated factors were associated with pneumococcal vaccine uptake.

## 4. Discussion

In this prospective cohort study of virologically suppressed PLWH, we found overall IRs of influenza and pneumococcal infections of 8.4 and 5.5 per 1000 person-years at risk, respectively. Though vaccine uptake for both vaccines increased from 2015 to 2020, the vaccine uptake in PLWH remained below recommendations. Factors significantly associated with vaccine uptake were previous episodes of bronchitis or pneumonia in the past 10 years, use of drugs to treat heart conditions, higher BMI, and longer time with HIV.

Seasonal influenza virus activity varies each year and may be influenced by several factors, including the circulating type and subtype of influenza virus, antigenic drift, the prevalence of other respiratory viruses, seasonal vaccine uptake, as well as local strategies to reduce viral spread [35]. Accordingly, we found incidences of confirmed influenza ranging between 0 (95% CI 0.0, 7.6) in 2020 and 18.0 in 2017 (95% CI: 8.2, 34.1) per 1000 person-years at risk from 2015 to 2020. These findings are in accordance with surveillance data from the Danish general population with a similarly higher influenza incidence in 2017 compared to previous seasons, reflecting a remarkably long and high-activity season in Denmark [36]. Similarly, in 2020 the incidence of influenza in the Danish general population (n = 5.875 million) was unusually low with a total of 46 positive PCR tests for influenza virus [37]. Similar trends were observed in other European countries, possibly reflecting the preventive efforts to curb the COVID-19 pandemic at the time [35]. Based on surveillance data from the Danish general population [36,38], the IRs of confirmed influenza in PLWH in the present study may be higher than in the background population, and this may indicate increased susceptibility to influenza virus in PLWH despite virological suppression, although increased testing of PLWH presenting with respiratory symptoms cannot be ruled out. Previous studies have reported IRs up to 46 per 1000 person-years [15,39,40]. However, these studies were either conducted in high HIV prevalence settings, based on the measurement of antibody titers, or included PLWH with unknown immune status, which could explain the differences in our findings. Though outcomes of influenza may be more severe, increased influenza virus susceptibility in PLWH regardless of cART has not uniformly been proven, underlining the need for further studies [40,41,42].

For overall pneumococcal infections and IPD, we found IRs of 5.5 and 0.3 per 1000 person-years, respectively, and all cases of IPD occurred in 2017. Based on surveillance data from the Danish general population [38,43], the corresponding IR of IPD in 2017 was 0.1 per 1000 person-years, which is slightly lower than our finding of IPD in PLWH. An IPD IR of 3.1 per 1000 person-years at risk has previously been reported in Danish PLWH. However, only 51% of the PLWH were on cART at the time of IPD, which could explain the differences in our findings [44]. Unfortunately, data regarding the incidence of non-invasive pneumococcal infections are unavailable for the Danish general population. Early cART initiation and related undetectable viral load, alongside the implementation of pneumococcal vaccinations in children and high-risk groups in many countries, including Denmark, has decreased the rates of pneumococcal infections in PLWH [45,46]. Yet, considerably higher incidence rates remain in PLWH with low CD4+ lymphocyte counts, detectable viral load, intravenous drug use, and in non-cART users [44,45,47]. However, five- and four-fold higher incidence rates of *S. pneumoniae* community-acquired pneumonia and IPD, respectively, have been reported in cART users compared to the general population [48]. As such, the high prevalence of pulmonary comorbidities in PLWH and HIV-induced chronic inflammation may predispose to an increased risk of pulmonary infections [7,42,46,48], such as influenza and pneumococcal infections, but little is known in a contemporary context of well-treated PLWH.

In the present study, only 22.6% of PLWH were influenza-vaccinated in all consecutive seasons, and 46.9% received no influenza vaccine at all. As such, influenza vaccine uptake was considerably below the European Centre for Disease Prevention and Control’s target of 75% coverage for seasonal influenza vaccination of vulnerable patients [49]. Similarly, 69% of PLWH had not received the pneumococcal vaccine, and our findings of low vaccine uptake in PLWH are in accordance with previous studies [28,50,51]. However, we did observe a peak in vaccine uptake for both vaccine types in 2020, possibly due to the increased awareness of vaccination caused by the COVID-19 pandemic. Similar trends in increasing vaccine uptakes were observed in other at-risk groups in Denmark, such as individuals over 65 years old [52]. Importantly, in April 2020 free vaccination with the 23-valent pneumococcal polysaccharide vaccine was implemented in Denmark for persons above 65 years with chronic illness, and persons below 65 years at risk of infection, including PLWH [26], a possible explanation for the high pneumococcal vaccine uptake in 2020 observed in our study.

In our study, factors associated with vaccine uptake after adjusting for age and sex were previous episodes of bronchitis or pneumonia in the past 10 years, use of drugs to treat heart conditions, higher BMI, and longer time with HIV. These findings suggest that PLWH with conceivably more contact with the health care system due to pulmonary or cardiovascular comorbidity may be more prone to receive influenza and pneumococcal vaccines. Accordingly, studies have found that influenza vaccine uptake increases with average number of prescriptions [53], respiratory illness in the prior year [53], at least one healthcare visit in the prior six months [53,54], as well as increasing age [28,50,51,54], and previous influenza vaccination [28].

Our study has some limitations. First, since the numbers of cases for both infections were low, we did not have the power to investigate risk factors associated with influenza and pneumococcal infections, and as such, we were unable to identify PLWH with potentially increased susceptibility to infection despite virological suppression. Second, we did not have a comparison group of individuals without HIV, and thus could not directly compare our results to the general population. Third, since we cannot account for untested cases, the reported incidences of confirmed influenza and pneumococcal infections in our study may be lower than the actual burden of these infections in PLWH. The strength of our study includes our large, well-described prospective cohort of virologically suppressed PLWH and low loss to follow-up alongside our nationwide microbiology (MiBa) and vaccination (DDV) databases.

## 5. Conclusions

In conclusion, we observed high incidences of confirmed influenza and pneumococcal infections in virologically suppressed PLWH. Importantly, the vaccine uptake in PLWH was below recommendations, and efforts should be made to increase awareness of recommendations to improve vaccine uptake among PLWH.

## Figures and Tables

**Figure 1 vaccines-13-00358-f001:**
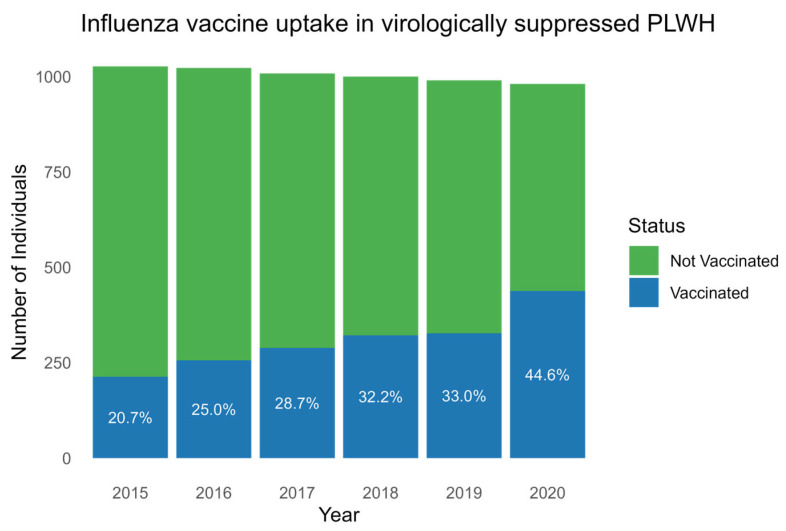
Influenza vaccine uptake each season between 2015 and 2020. Boxplot showing the number of PLWH (y-axis) with influenza vaccine uptake (blue) and without influenza vaccine uptake (green) in the years 2015 to 2020 (x-axis). Percentages of vaccine uptake for each year are written in white. Abbreviation: PLWH: people living with HIV.

**Figure 2 vaccines-13-00358-f002:**
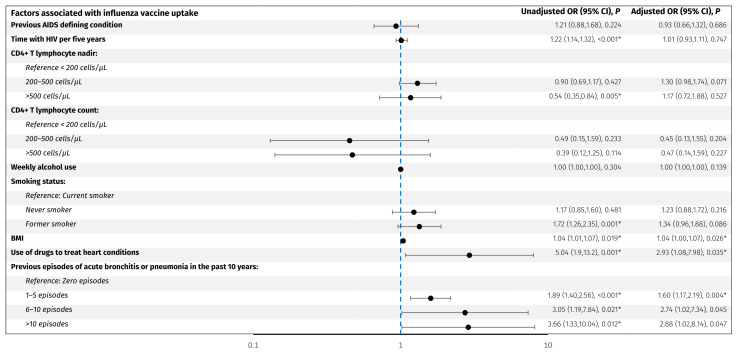
Predictors associated with influenza vaccine uptake. The forest plot shows odds ratios adjusted for age and sex for factors associated with influenza vaccine uptake in PLWH. The y-axis presents the investigated variables, and the x-axis (logarithmic scale) represents odds ratios. The blue line represents an odds ratio of 1. *p*-values < 0.05 (*) were considered statistically significant. Abbreviations: BMI: body mass index. CI: confidence interval. PLWH: people living with HIV.

**Figure 3 vaccines-13-00358-f003:**
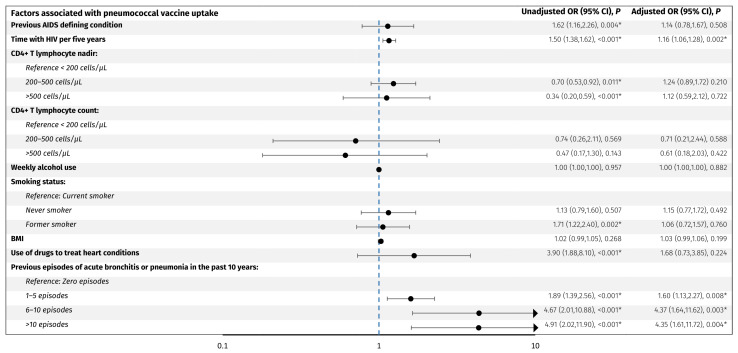
Predictors associated with pneumococcal vaccine uptake. The forest plot shows odds ratios adjusted for age and sex for factors associated with pneumococcal vaccine uptake in PLWH. The y-axis presents the investigated variables, and the x-axis (logarithmic scale) represents odds ratios. The blue line represents an odds ratio of 1. *p*-values < 0.05 (*) were considered statistically significant. Abbreviations: BMI: body mass index. CI: confidence interval. PLWH: people living with HIV.

**Table 1 vaccines-13-00358-t001:** Demographics, HIV-related characteristics, and vaccine uptake for all 1031 PLWH.

General characteristics		
Age, years, median (IQR)		50.7 (15.2)
Male, n (%)		878 (84.7%)
Smoking status, n (%)	Current smoker Former smoker Never smoker	291 (28.2%) 357 (34.6%) 347 (33.7%)
Pack years in current and previous smokers, mean (SD)		23.0 (22.3)
Weekly alcohol consumption in units, mean (SD)		10.3 (12.4)
BMI (kg/m^2^), mean (SD)		25.0 (4.1)
Use of drugs to treat heart conditions, yes, n (%)		32 (3.1%)
Acute bronchitis or pneumonia in the past 10 years, n (%)	1–5 episodes 6–10 episodes >10 episodes	256 (24.8%) 24 (2.3%) 22 (2.1%)
HIV characteristics		
CD4+ T cell count, cells/µL, mean (SD)		724.2 (282)
CD4+ T cell count, cells/µL, category, n (%)	<200 200–500 >500	15 (1.5%) 194 (18.8%) 816 (79.2%)
CD4+ T cell nadir, cells/µL, mean (SD)		252.1 (181)
CD4+ T cell nadir, category, n (%)	<200 200–500 >500	416 (40.5%) 485 (47.0%) 107 (10.4%)
Previous AIDS-defining conditions, n (%)		182 (17.7%)
Use of cART, n (%)		1020 (98.9%)
Time with HIV, years, mean (SD)		14.5 (9.0)
Influenza vaccination		
Influenza-vaccinated in any season, n (%)		548 (53.2%)
Influenza-vaccinated in all seasons, n (%)		124 (22.6%)
Influenza-vaccinated in one season, n (%)		150 (27.4%)
Influenza-vaccinated in two seasons, n (%)		73 (13.3%)
Influenza-vaccinated in three seasons, n (%)		74 (13.5%)
Influenza-vaccinated in four seasons, n (%)		46 (8.4%)
Influenza-vaccinated in five seasons, n (%)		81 (14.8%)
No influenza vaccination, n (%)		483 (49.9%)
Pneumococcal vaccination		
Pneumococcal-vaccinated in 2015, n (%)		4 (1.2%)
Pneumococcal-vaccinated in 2016, n (%)		9 (2.8%)
Pneumococcal-vaccinated in 2017, n (%)		10 (3.1%)
Pneumococcal-vaccinated in 2018, n (%)		10 (3.1%)
Pneumococcal-vaccinated in 2019, n (%)		13 (4.1%)
Pneumococcal-vaccinated in 2020, n (%)		235 (73.4%)
Pneumococcal-vaccinated in 2021, n (%)		39 (12.2%)
No pneumococcal vaccination, n (%)		711 (69.0%)

Abbreviations: BMI: body mass index, cART: combination antiretroviral therapy, IQR: interquartile range, SD: standard deviation. PLWH: people living with HIV.

**Table 2 vaccines-13-00358-t002:** The incidence rate of confirmed influenza per 1000 person-years at risk each season between 2015 and 2020. An influenza season was defined as the period from 1 October to 31 March the following year. Abbreviations: IR: incidence rate. CI: confidence intervals.

Seasons	Influenza IR per 1000 Person-Years at Risk	95% CI
2015	5.9	1.2, 17.1
2016	4.0	0.5, 14.3
2017	18.0	8.2, 34.1
2018	15.6	6.8, 30.8
2019	6.1	1.3, 17.8
2020	0.0	0.0, 7.6
2015–2020	8.4	4.4, 12.3

## Data Availability

The data are not publicly available due to privacy or ethical restrictions. The data that support the findings of this study are available from the corresponding author upon reasonable request.

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
