# Peer review of "Incidence of Confirmed Influenza and Pneumococcal Infections and Vaccine Uptake Among Virologically Suppressed People Living with HIV"

_vaccines, 2025, doi:10.3390/vaccines13040358_

Round 1
Reviewer 1 Report
Comments and Suggestions for Authors
This manuscript can be published. However, I have found some special remarks and comments that should be considered to make the text more readable.
Major remarks:
Throughout the entire manuscript, flu is confused with the influenza virus. This needs to be corrected. Below are listed selected examples.:
Line 21-22: the sentence: “Influenza and Streptococcus pneumoniae are common vaccine-preventable diseases to which people living with HIV (PLWH) may be more susceptible.” should be overwritten because Streptococcus pneumoniae is not a disease but an infectious agent that can be responsible for infections,
Line 55: influenza is a disease caused by the influenza virus, so the sentence in this line should also be corrected.
Minor remarks:
Line 27: the abbreviation “(IR)” is repeated; therefore, delete it,
Line 51: delete “(S. pneumoniae)” because, in the next part of the text, the notation of the species name “S. pneumoniae” is correctly used for the species name of this bacteria,
Line 206: change the capital “N” letter into “n” lowercase,

Reviewer 2 Report
Comments and Suggestions for Authors
The authors investigated the incidences of influenza and pneumococcal infections and factors associated with vaccine uptake in a population of PLWH, and found high infection incidences but low vaccine uptakes. Although this study has some limitations, such as a small case number and no comparison group, it can still provide some useful information to this field. However, there are some points need to be improved.
Table 1: some % symbols are missing from the third column.
Section 3.2: A table can show all these data more clearly and comparable, which makes this section easier to read and understand. Please also make the decimal places of all CI values consistent (use either 1 or 2).
Figure 2 and 3: Does the x axis represent adjusted or unadjusted odds ratios? Anyway, I suggest that unadjusted odds ratios should also be listed like adjusted odds on the right side, so we can compare them. The investigated variables on y axis should be formatted better so that we can distinguish the subtitle and item. In addition, “Odds ratios and 95% confidence intervals in brackets” but odds ratios look not in brackets.
Round 2
Reviewer 1 Report
Comments and Suggestions for Authors
I have no additional remarks. The manuscript is well-written and should be published.